# The Effectiveness of Clinician-Led Community-Based Group Exercise Interventions on Health Outcomes in Adults with Type 2 Diabetes Mellitus: A Systematic Review and Meta-Analysis

**DOI:** 10.3390/ijerph21050601

**Published:** 2024-05-07

**Authors:** Liam White, Morwenna Kirwan, Vita Christie, Lauren Hurst, Kylie Gwynne

**Affiliations:** 1Faculty of Medicine, Health and Human Sciences, Macquarie University, Talavera Road, North Ryde, NSW 2109, Australia; liamtwhite8@gmail.com (L.W.); morwenna.kirwan@mq.edu.au (M.K.); lauren.hurst1@hotmail.com (L.H.); 2Djurali Centre for Aboriginal and Torres Strait Islander Health Research, Heart Research Institute, Eliza Street, Newtown, NSW 2042, Australia; kylie.gwynne@hri.org.au; 3DVC Indigenous Office, University of New South Wales, High Street, Sydney, NSW 2052, Australia

**Keywords:** type 2 diabetes mellitus, community-based, clinician-led, group exercise, glycemic control, physical fitness, anthropometric, health outcomes

## Abstract

This systematic review and meta-analysis evaluated the combined effects of clinician-led and community-based group exercise interventions on a range of health outcomes in adults with type 2 diabetes mellitus. Our literature search spanned Medline, Scopus, PubMed, Embase, and CINAHL databases, focusing on peer-reviewed studies published between January 2003 and January 2023. We included studies involving participants aged 18 years and older and articles published in English, resulting in a dataset of eight studies with 938 participants. Spanning eight peer-reviewed studies with 938 participants, the analysis focused on the interventions’ impact on glycemic control, physical fitness, and anthropometric and hematological measurements. Outcomes related to physical fitness, assessed through the six-minute walk test, the 30 s sit-to-stand test, and the chair sit-and-reach test, were extracted from five studies, all of which reported improvements. Anthropometric outcomes from seven studies highlighted positive changes in waist circumference and diastolic blood pressure; however, measures such as body mass index, systolic blood pressure, weight, and resting heart rate did not exhibit significant changes. Hematological outcomes, reviewed in four studies, showed significant improvements in fasting blood glucose, triglycerides, and total cholesterol, with glycemic control evidenced by reductions in HbA1c levels, yet LDL and HDL cholesterol levels remained unaffected. Ten of the fifteen outcome measures assessed showed significant enhancement, indicating that the intervention strategies implemented may offer substantial health benefits for managing key type 2 diabetes mellitus-related health parameters. These findings in combination with further research, could inform the refinement of physical activity guidelines for individuals with type 2 diabetes mellitus, advocating for supervised group exercise in community settings.

## 1. Introduction

Diabetes affects more than half a billion individuals globally, with type 2 diabetes mellitus comprising 90% to 95% of these cases [1]. This prevalence, representing nearly one in ten adults, is increasing and poses a significant threat to the health and well-being of people, impacting individuals, families, and societies [1]. 

Physical activity, alongside nutritional and medical therapies, is critical for managing type 2 diabetes mellitus [2]. Exercise, a specific category of physical activity, involves activities that improve strength, endurance, agility, balance, and flexibility, all of which are beneficial for type 2 diabetes mellitus patients. These benefits extend beyond physical health, positively affecting the psychological and cognitive aspects of health [2]. Current guidelines advise adults aged 18–64 with type 2 diabetes mellitus to undertake at least 150 min of moderate-to-vigorous intensity exercise weekly and to participate in resistance training sessions at least twice a week [2]. Despite such guidelines, over 1.4 billion adults globally fall short of meeting these physical activity recommendations, regardless of their type 2 diabetes mellitus status [3].

Researchers have investigated several physical activity intervention techniques to support people with type 2 diabetes mellitus to be more active. Clinician-led facility-based fitness training is one such tactic, and it has the potential to enhance glycemic management and other cardiovascular risk factors of type 2 diabetes mellitus [4,5,6]. These interventions are frequently resource-intensive, only accessible in large cities, and their long-term viability is uncertain [7]. Other methods to encourage physical activity in type 2 diabetes mellitus adults include individual-based treatments, medication use, and behavior modification. It can be difficult to persuade people with type 2 diabetes mellitus to embrace behavior change with only short visits to their GP [7]. These self-management techniques are also only moderately effective in the near term, and long-term evaluations are frequently relatively few [8]. Furthermore, those with low incomes, low levels of education, limited access to healthcare, and linguistic and cultural hurdles may find these types of treatments to be inaccessible [8].

The burgeoning prevalence of type 2 diabetes mellitus necessitates an expansion of existing intervention strategies to manage the disease effectively. The synthesis of clinician-led and community-based exercise interventions presents a promising hybrid model, leveraging the structured guidance of healthcare professionals with the accessibility of community settings. Recognizing the potential of this integrated approach could be instrumental in shaping future health policies and guidelines that seek to amplify the reach and impact of type 2 diabetes mellitus management strategies. Community-based exercise interventions might overcome the limitations of facility-based and individual approaches by providing culturally relevant health education. Facility-based interventions are administered in controlled, institutional environments such as hospitals or clinics, while community-based interventions take place within local settings, utilizing area resources and engaging community members, potentially increasing adherence to self-management practices [9,10]. The World Health Organization advocates for such interventions to promote physical activity among people with type 2 diabetes mellitus [11]. Updates to physical activity guidelines now recommend clinician-led exercise as a beneficial strategy [12]. A systematic review in 2018 indicated that supervised aerobic and resistance training yields better health outcomes than unsupervised activities [13]. Studies combining community-based and clinician-led exercises have shown health benefits [14,15], with some effects persisting for up to twelve months post-intervention [16].

This study evaluated the combination of clinician-led and community-based exercise interventions of adults with type 2 diabetes mellitus through a systematic review and meta-analysis. In this context, a “clinician” is a qualified health worker who delivers services in community settings. The study design anticipated sufficient quantitative data to include a meta-analysis. Unlike previous studies conducted in a workplace or traditional clinical settings, this research investigates the effectiveness of group exercise interventions for adults with type 2 diabetes mellitus implemented in community-based settings such as recreation centers, local facilities, and community centers. To our knowledge, this is the first systematic review targeting the efficacy of supervised group exercise interventions in community settings for enhancing health outcomes in type 2 diabetes mellitus adults.

## 2. Materials and Methods

### 2.1. Study Design

This systematic review and meta-analysis was conducted according to the Preferred Reporting Items for Systematic Reviews and Meta-Analyses (PRISMA) statement [17] and was registered with the International Prospective Register of Systematic Reviews (PROSPERO: ID no. CRD42023363265).

In adherence to the Preferred Reporting Items for Systematic Reviews and Meta-Analyses (PRISMA) guidelines [17], the systematic search and selection process for relevant studies was conducted across multiple databases. A comprehensive search yielded a total of 693 studies from CINAHL, Scopus, Embase, Medline, and PubMed. Following the removal of 264 duplicates, 429 abstracts were screened. Of these, 415 studies were excluded based on exclusion criteria, leaving 14 full-text articles to be assessed for eligibility. Reasons for exclusion at this stage included the absence of outcome data reported as mean and standard deviation and the lack of integration of both clinician-led and community-based approaches in the interventions. Ultimately, 8 studies met all criteria and were included in the systematic review and meta-analysis.

### 2.2. Search Strategy and Data Sources

The search strategy was developed in consultation with two senior health researchers (MK, KG) and a health research librarian. A comprehensive literature search across Medline, Scopus, Pubmed, Embase, and CINAHL databases for peer-reviewed studies published from January 2003 until January 2023 was conducted. We employed the following search strings to gather relevant data: “(Type 2 Diabetes OR Diabetes) AND (Clinician Led OR Supervised) AND (Community-based OR Community) AND (Exercise OR Physical Activity OR Fitness OR Outcome Measures) AND (Management)”. These search parameters were refined to include subjects aged 18 years and older and articles published in the English language (Appendix A).

Initial identification of titles and abstracts was independently performed by two authors (LW and MK), with disparities resolved through discussion or consultation with a third reviewer (KG) if required. Full texts of potentially relevant studies were then retrieved and assessed for eligibility. The final inclusion of articles was determined by checking the references of selected studies for additional relevant literature. All search results were systematically organized using Microsoft Excel (Version 16.84).

For this review, we adopted a PICO framework focusing on:

Participants/Population: Adults (18 years or older) involved in clinician-led, community-based exercise programs for managing type 2 diabetes mellitus.

Intervention(s)/Exposure(s): Eligible studies were those conducted in high-income countries—specified regions (Australia, New Zealand, Canada, UK, and Europe), with interventions predominantly based on PA (over 50%), targeting adults with pre-existing type 2 diabetes mellitus, using community-based settings, and overseen by qualified clinicians, presenting quantitative studies of original data in peer-reviewed journals.

Comparator(s)/Control: Participants receiving standard care without the specified clinician-led, community-based exercise.

Main Outcome(s): Measurable pre- and post-health outcomes related to PA, including weight loss, BMI changes, waist-to-hip ratios, HbA1c levels, and six-minute walk test (6MWT) improvements.

Additional Outcome(s): Compliance with PA programs, clinician experience, and details on the interventions’ setting and delivery.

### 2.3. Inclusion and Exclusion Criteria

This review included full-text, published, peer-reviewed literature with original outcome data that reported on the effectiveness of clinician-led, community-based, group exercise interventions for the management of type 2 diabetes mellitus. The inclusion criteria included lifestyle interventions where exercise was a purposeful, structured, and required part of the intervention and conducted in high-income countries (Australia, New Zealand, Canada, UK, and Europe). Studies published 2003–2023 were included to obtain contemporary evidence.

Studies were omitted if they were not peer-reviewed, published prior to 2003, were not published in English, and did not feature interventions primarily based in the community. Exclusions also applied to research not primarily focused on physical activity, those that involved minors, or adults with diabetes types other than type 2 diabetes mellitus where specific data were not differentiated. Additional exclusion criteria encompassed studies conducted exclusively in clinical environments, those without the oversight of a professionally trained clinician, and research relying solely on qualitative data sources.

### 2.4. Data Extraction and Management

Following the study selection, data were extracted into a Microsoft Excel spreadsheet. This data included the title, authors, publication year, study design, location, clinician type, inclusion and exclusion criteria, participant details, type of physical activity intervention, outcome measures, and results.

### 2.5. Meta-Analysis

For the synthesis of results, we conducted a meta-analysis of comparable outcome measures, including glycemic control, physical fitness outcomes, and anthropometric measurements, employing the mean difference pooled using a random effects model [18] in IBM SPSS Statistics (Version 28). The effect size was calculated for each relevant outcome measure at a 95% confidence interval. Heterogeneity was evaluated using the I-squared statistic. The included studies’ effect estimations were represented graphically by forest plots. A *p*-value of <0.05 was considered statistically significant. Revman Web was used to display figures represented within this systematic review [19].

### 2.6. Risk of Bias Assessment

The risk of bias in included studies was appraised using the EPHPP tool [20] by two authors (LW and LH) independently. The EPHPP tool was selected as a suitable tool for assessment of quantitative public health research. Any conflicting evaluations were discussed until a consensus was reached. The final inclusion of studies in the review depended on achieving at least a medium-quality rating.

## 3. Results

### 3.1. Study Selection Process

This systematic review and meta-analysis was conducted in strict accordance with the PRISMA guidelines [17], which dictate standards for reporting such research. The review started with a detailed search across several databases, namely CINAHL, Scopus, Embase, Medline, and PubMed. This initial search yielded 693 studies. From these, 264 duplicates were identified and removed, leaving 429 for abstract screening. After careful consideration, 415 studies were excluded because they did not meet the specific criteria set out for this research.

The eligibility of the remaining 14 full-text articles was closely examined. Five of these articles were excluded for reasons including not reporting outcomes in the form of mean and standard deviation, or because they did not feature both clinician-led and community-based intervention approaches. Consequently, eight studies were selected for inclusion in the final review and analysis. This work was also officially registered with the International Prospective Register of Systematic Reviews (PROSPERO) [21] under the ID CRD42023363265, ensuring that the process was systematically planned and recorded. This has been displayed in our PRISMA flowchart (Figure 1).

### 3.2. Study Characteristics

The eight studies encompassed a total of 938 participants and explored 15 health outcomes to ascertain the impact of clinician-led and community-based group exercise programs on adults with type 2 diabetes mellitus. We have detailed the characteristics of these studies in Table 1, which includes participant demographics, intervention types, durations, and settings. Requests for additional information were made to the authors of two studies [22,23].

### 3.3. Quality of Included Studies

Quality assessment using the EPHPP framework indicated that five studies were rated as “strong” overall, with the remaining three rated as “medium” (Table 2).

Most of the studies recruited participants through diabetes management clinics or patient databases. Among them, one study was conducted as a randomized controlled trial [23], while the other seven were pre-post studies. Confounding variables were identified in four of the studies [14,23,25,27]. It is noteworthy that all studies employed data collection methods recognized for their validity and reliability. However, there was a lack of clarity in the reporting of participant withdrawals and dropouts in two of the studies [14,15].

### 3.4. Cardiometabolic Health Indicators

Anthropometric measurements, as well as heart rate and blood pressure, were focal outcomes in seven studies, as depicted in Figure 2 [14,15,23,24,25,27,28]. A notable and statistically significant reduction emerged in both waist–hip circumference (CI: −4.27 to −1.70) and diastolic blood pressure (CI: −6.53 to −2.38). Trends suggesting improvement were seen in BMI (CI: −22.79 to 3.46) and systolic blood pressure (CI: 13.99 to 2.19), though these changes did not reach statistical significance. No associations were noted for either weight (CI: −2.17 to 1.54) or resting heart rate (CI: −1.20 to 1.46).

### 3.5. Physical Fitness and Functional Capacity

A suite of tests designed to evaluate physical and functional fitness, presented in Figure 3 [14,15,23,26,28], indicated significant improvements. These were quantified in the 6 min walk test (6MWT) (CI: 42.38 to 88.42), the 30 s sit-to-stand test (STS 30) (CI: 2.92 to 4.67), and the chair sit-and-reach test (CI: 2.68 to 5.52).

### 3.6. Glycemic and Lipid Profiles

The efficacy of the interventions on glycemic control and lipid metabolism was captured through hematological measures in four studies, as shown in Figure 4 [23,24,27,28]. There were statistically significant improvements in HbA1c (CI: −0.94 to −0.30), fasting blood glucose (CI: −26.37 to −9.73), triglycerides (CI: −39.95 to −18.48), and total cholesterol (CI: −26.41 to −2.89). LDL cholesterol presented a favorable trend (CI: −22.79 to 3.46), although not statistically significant. HDL cholesterol did not exhibit significant change post-intervention (CI: −3.33 to 4.17).

## 4. Discussion

This systematic review supports the assertion that a dual strategy encompassing clinician-led and community-based group exercise interventions can improve important health outcomes in adults with type 2 diabetes mellitus. Out of fifteen health outcomes assessed through meta-analysis, nine demonstrated statistically significant improvements associated with the intervention. This resonates with findings from prior systematic reviews, which similarly concluded that exercise interventions delivered in community or clinical settings are efficacious for type 2 diabetes mellitus management [7,13].

Cardiometabolic Health Indicators: Our analysis identified significant enhancements in waist circumference and diastolic blood pressure among the intervention group. The implications of these improvements are noteworthy considering the pivotal role these indicators play in forecasting glycemic control within the type 2 diabetes mellitus population [28,29]. Although no substantial association was found between the interventions and BMI, body weight, resting heart rate, or systolic blood pressure, these parameters have been reported to improve in earlier interventions as per previous reviews [7,13].

Physical Fitness and Functional Capacity: The interventions included in our meta-analysis also significantly improved cardiovascular fitness, lower body strength, and flexibility. These findings hold particular importance because individuals with type 2 diabetes mellitus are prone to a gradual decline in physical function due to aging, which often leads to muscle atrophy and an increase in fat mass. These changes can substantially restrict mobility and physical function [30]. Our review indicates that intervention periods as brief as 2 to 3 months are capable of significantly enhancing the functional fitness of adults with type 2 diabetes mellitus.

Glycemic and Lipid Profiles: Hematological measures, including HbA1c, fasting blood glucose, triglycerides, and total cholesterol, also showed significant improvements post-intervention. Notably, despite a shorter intervention span of 8 weeks, the study by Akinci et al. [23] reported changes in mean values comparable to those from studies with longer durations, each extending beyond 3 months [24,27,28]. This observation is particularly striking as HbA1c levels reflect an individual’s average blood glucose control over a three-month period [31], suggesting that even brief interventions can be beneficial. However, due to the nature of HbA1c, long-term follow-up in future studies may be required to understand the full impact of these interventions [24].

Research that evaluates the health benefits of physical activity for older adults with type 2 diabetes mellitus yields findings of considerable importance. A concerning trend highlighted by current research is the lower fitness levels observed in individuals with type 2 diabetes mellitus as compared to non-diabetic individuals, a disparity that is worsened with aging [32]. Improvements in physical and functional fitness are crucial for this group, as they are intimately linked with reductions in cardiovascular risks and improvements in insulin sensitivity [33]. Nonetheless, the task of evaluating the clinical importance of such physical and functional fitness improvements is complex. Quantifying the clinical impact of the physical and functional fitness improvements noted in our systematic review is difficult due to limited data on what constitutes a clinically significant change for this population [34].

To address this, validated criterion standards, though not specific to type 2 diabetes mellitus, offer valuable benchmarks for the fitness levels necessary for older adults to remain independent [35]. Our systematic review incorporated two studies [14,15] that used these benchmarks to assess participants’ baseline fitness and subsequent improvements. These benchmarks were critical for quantifying participants’ fitness levels in relation to the standards needed for independence with aging. The findings from Kirwan et al. [14,15] were encouraging, showing a considerable number of participants reaching or surpassing the target fitness levels after the intervention. These findings bolster the case for future research to utilize these benchmarks in community-based, clinician-led group exercise programs, which would allow for a more nuanced interpretation of changes in the physical and functional fitness among individuals with type 2 diabetes mellitus.

In the current discourse on geriatric health, the significance of functional fitness emerges as a critical factor, particularly within the demographic contending with type 2 diabetes mellitus. Maintaining an adequate level of functional fitness is instrumental in diminishing fall risks and fostering the capacity for independent living, which in turn exerts a substantial influence on the quality of life [36]. For individuals navigating the complexities of type 2 diabetes mellitus management, preserving their independence and the ability to conduct daily living activities constitutes a fundamental health objective [32]. A deficiency in effective, targeted interventions may result in a trajectory that culminates in dependency, precipitating substantial long-term economic burdens on healthcare systems. Complicating this issue is the prevalence of physical inactivity among individuals with type 2 diabetes mellitus, which further exacerbates the risk of functional decline [37,38].

The studies included in this review showed favorable health outcomes for participants; however, these studies varied in duration from 8 weeks to 9 months. It remains to be seen whether the short-term benefits translate into long-term health improvements over the years. Longitudinal studies are needed to determine if the observed benefits persist and to investigate methods to encourage sustained engagement with these exercise programs.

The cost-effectiveness of combined clinician-led and community-based interventions also warrants further exploration. Community-based interventions are often perceived as more economical compared to their clinic-based counterparts. Nevertheless, additional financial considerations such as staffing, program development, and the upkeep of facilities must not be overlooked. A recent systematic review has summarized the existing economic evaluations of physical activity interventions specifically in the context of type 2 diabetes mellitus management [39]. The findings are encouraging, indicating that such interventions are generally a sound investment—four out of ten interventions were deemed cost-saving, while six were considered cost-effective, and two displayed favorable cost-utility characteristics. Future research should, therefore, extend to a thorough cost-effectiveness and cost-utility assessment of combined community-based and clinician-led group exercise interventions. Such inquiry is essential to assess the practicality and potential for broader application within diverse healthcare systems.

This research acknowledges several limitations that warrant caution in interpreting the findings. Primarily, the recruitment of participants from referred sources such as clinics may introduce bias, as opposed to random selection from a representative target population which could offer a more balanced perspective. Furthermore, the reporting of participant adherence was often absent or noted to be moderate at best. For instance, Higgs et al. observed a dropout rate of approximately 40 percent before the follow-up measures could be taken [24]. Additionally, the participant gender ratio in many included studies was skewed, potentially affecting the extrapolation of results to the broader type 2 diabetes mellitus community, there are no molecular data or experimentally derived data and it is not clear whether these interventions can be applied to different geographical locations and different people. Despite these constraints, efforts were made to design this review to maximize the translational potential of the findings.

In terms of strengths, this review’s robust sample size of 938 participants enhances the reliability of the conclusions drawn. The deliberate inclusion of anthropometric, functional, and hematological measures offers a comprehensive view of the multifaceted impacts that clinician-led and community-based exercise interventions may have on adults with type 2 diabetes mellitus. Additionally, the selection of studies from countries with analogous cultures—such as Australia, New Zealand, Canada, the UK, and Europe—was intended to ensure participant homogeneity.

The insights garnered from this systematic review could inform refinements to physical activity guidelines tailored for individuals with type 2 diabetes mellitus, advocating for community-based, clinician-led exercise modalities. Notably, current Australian exercise recommendations for type 2 diabetes mellitus management mirror those for the general population, with specific guidance on blood glucose management during physical activity [2,40]. However, our review does not conclusively favor a particular community setting or clinician type, which may stem from the diverse community structures and clinician roles across the various countries of the included studies. This area might benefit from further investigative efforts.

## 5. Conclusions

In conclusion, our analysis of eight studies presenting quantitative data corroborates the expanding evidence that both community-based and clinician-led group exercise interventions can positively influence health outcomes in individuals with type 2 diabetes mellitus. Yet, the need for additional research is evident, especially regarding the interventions’ effectiveness within culturally and linguistically diverse groups. By addressing this gap, we can move towards resolving persisting uncertainties and creating customized interventions that address the distinct needs of these communities.

## Figures and Tables

**Figure 1 ijerph-21-00601-f001:**
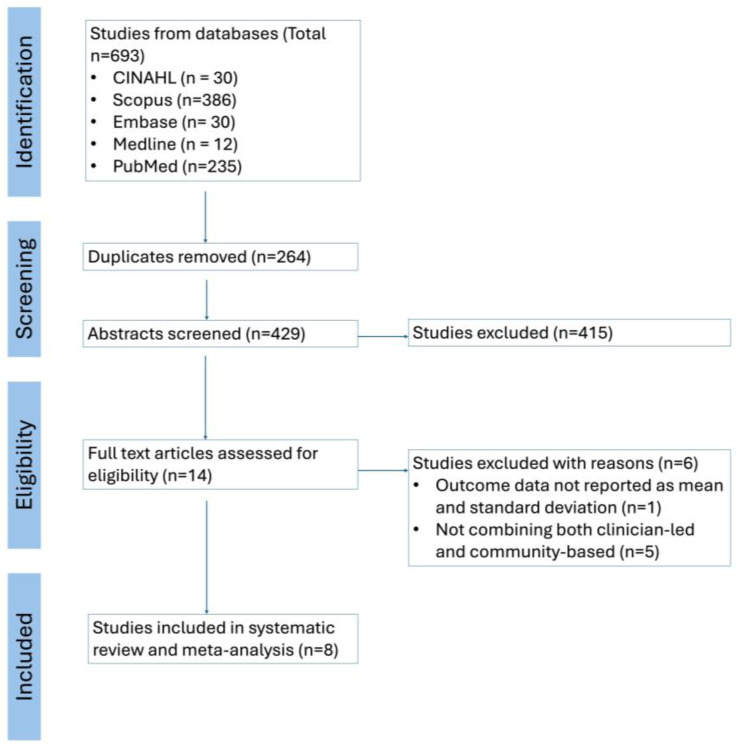
PRISMA (Preferred Reporting Items for Systematic Reviews and Meta-Analyses) flow diagram illustrating the selection process for studies included in the systematic review and meta-analysis.

**Figure 2 ijerph-21-00601-f002:**
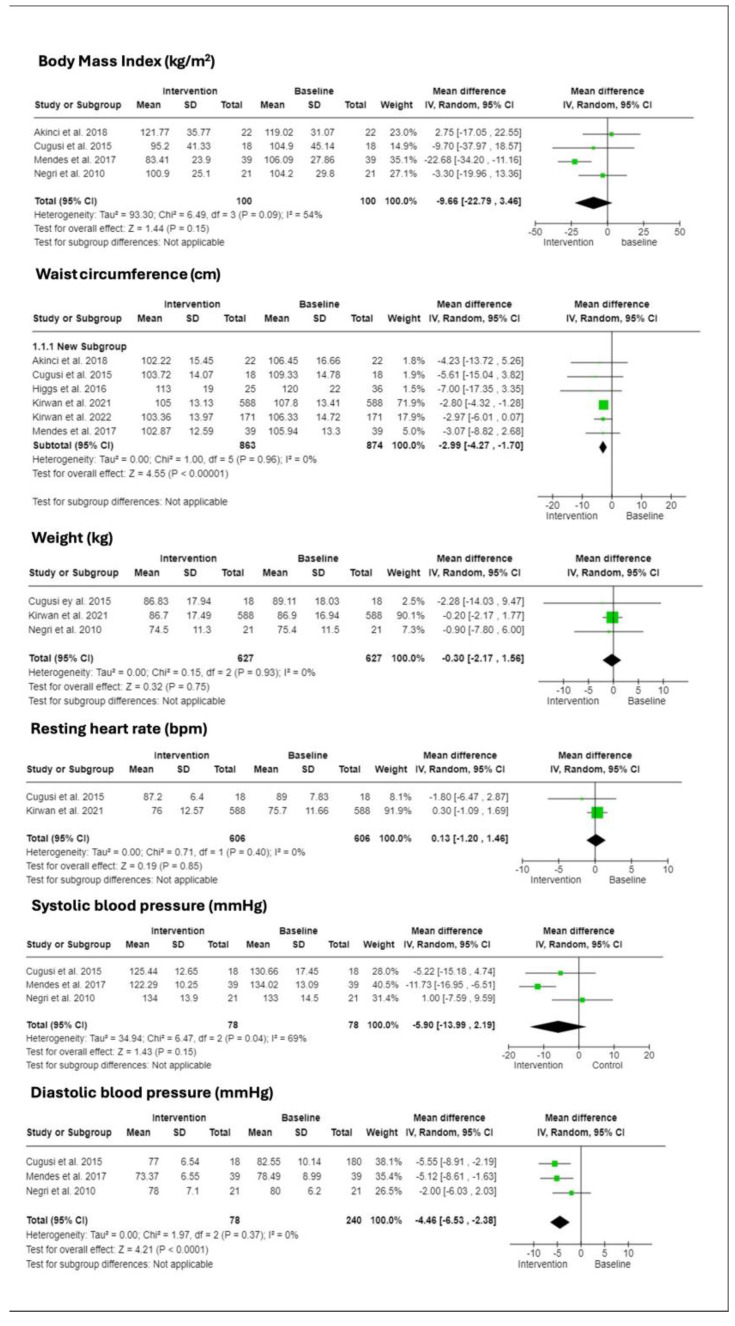
Forest plots depicting the effects of interventions on cardiometabolic parameters: body mass index (BMI, kg/m^2^ where kg = kilograms, m^2^ = square meters), waist circumference (cm, where cm = centimeters), weight (kg), resting heart rate (bpm, where bpm = beats per minute), systolic blood pressure and diastolic blood pressure, both measured in mmHg (millimeters of mercury). Each plot displays mean values, standard deviations (SD), total participant counts, and study-specific weights. Mean differences are shown with 95% confidence intervals (CI), calculated using a random effects model. Heterogeneity is quantified by I^2^ and tau-squared (τ^2^) values. Studies referenced: Kirwan et al., 2022 [14], Kirwan et al., 2021 [15], Akinci et al., 2018 [22], Mendes et al., 2017 [23], Higgs et al., 2016 [24], Cugusi et al., 2015 [26], Negri et al., 2010 [27].

**Figure 3 ijerph-21-00601-f003:**
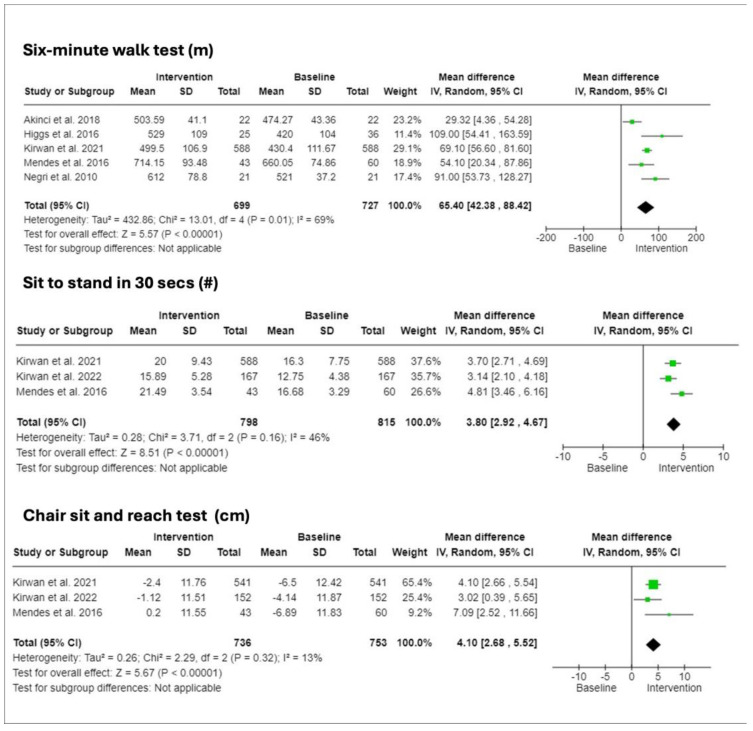
Forest plots of intervention effects on physical and functional fitness measures. Top panel: Six-Minute Walk Test (6MWT) measured in meters (m), evaluating walking distance. Middle panel: Sit-to-Stand in 30 Seconds Test (STS 30), counting the number of repetitions. Bottom panel: Chair Sit-and-Reach Test measured in centimeters (cm), assessing reach and flexibility. Each plot includes data on mean and standard deviation (SD) for intervention and baseline, total study weight, and the mean difference with 95% confidence intervals (CI). Heterogeneity across studies is quantified by I^2^ statistics and Tau^2^ values. Notably, significant improvements in physical performance measures are observed following the intervention, as indicated by the mean differences and confidence intervals. Studies referenced: Kirwan et al., 2022 [14], Kirwan et al., 2021 [15], Akinci et al., 2018 [22], Higgs et al., 2016 [24], Mendes et al., 2016 [25], Negri et al., 2010 [27].

**Figure 4 ijerph-21-00601-f004:**
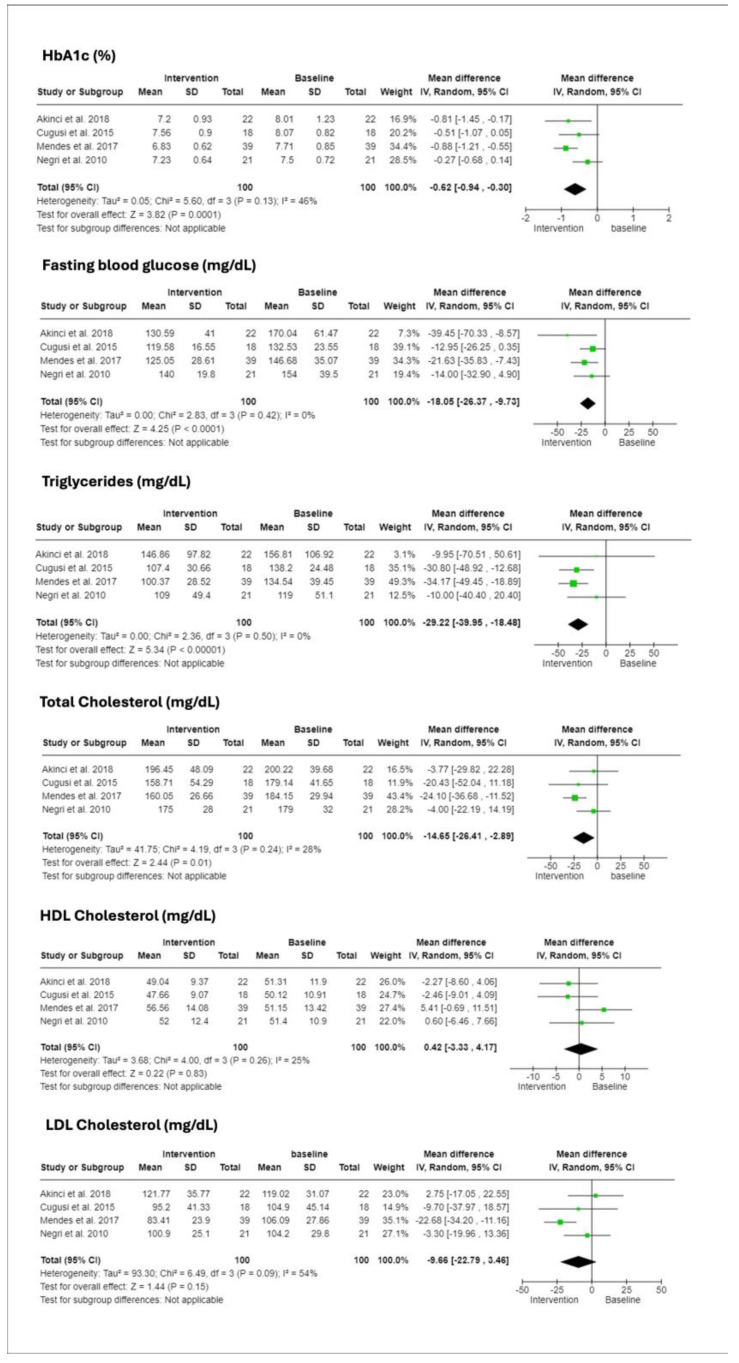
Forest plots displaying the effects of interventions on glycemic and lipid profiles measured in milligrams per deciliter (mg/dL). The plots are organized by biomarker: hemoglobin A1c (HbA1c), Fasting blood glucose, triglycerides, total cholesterol, high-density lipoprotein cholesterol (HDL), and low-density lipoprotein cholesterol (LDL). Each plot provides the mean and standard deviation (SD) at baseline and post-intervention, total sample size, mean difference with 95% confidence intervals (CI), and the weight of each study in the meta-analysis. Heterogeneity is quantified using I^2^ statistics, indicating the percentage of total variation across studies due to heterogeneity rather than chance. The intervention effect is assessed with a random effects model, showcasing significant changes in each biomarker following the intervention. Studies referenced: Akinci et al., 2018 [22], Mendes et al., 2017 [23], Cugusi et al., 2015 [26], Negri et al., 2010 [27].

**Table 1 ijerph-21-00601-t001:** Individual study characteristics.

Study Authors and Date	Participants		Intervention Duration	Intervention Type	Setting	Supervising Clinician
	N (% Female)	Age (y), Mean (SD)				
Kirwan et al., 2022 [14]	171 (68)	71 (5.6)	8 weeks	Twice weekly synchronous group exercise sessions including a dynamic warm-up and cooldown, aerobic, resistance, balance, and flexibility exercises	Online—participant’s home	Accredited Exercise Physiologist
Kirwan et al., 2021 [15]	588 (52)	69.8 (5.6)	8 weeks	Twice weekly group exercise sessions including a dynamic warm-up and cooldown, aerobic, resistance, balance, and flexibility exercises	Varied locations including public gyms, community halls, and private clinics	Accredited Exercise Physiologist
Akinci et al., 2018 [22]	22 (81.8)	53.59 (6.02)	8 weeks	Three times a week of group based aerobic and resistance exercises.	University Faculty of Health Sciences	Physiotherapist
Mendes et al., 2017 [23]	39 (51.3)	62.05 (6.14)	9 months	Three times a week of group based aerobic, resistance, agility/balance, and flexibility exercises	Community sports complex	Exercise professional and nurse
Higgs et al., 2016 [24]	36 (58)	62 (11)	12 weeks	Twice weekly group sessions comprising education on a variety of health-related topics and exercise. Which Included an aerobic warmup, followed by resistance, aerobic and flexibility exercises	University fitness center	Physiotherapist and student physiotherapists
Mendes et al., 2016 [25]	43 (51)	62.51 (5.92)	9 months	Three times a week consisting of a mix of aerobic, resistance and balance training	Community sports complex	Exercise professionals
Cugusi et al., 2015 [26]	18 (0)	52.2 (9.28)	12 weeks	Three times a week consisting of a group aquatic- based exercise program. Which included aerobic, resistance and stretching exercises.	Aquatic center	NR
Negri et al., 2010 [27]	21 (NR)	65.7 (4.9)	4 months	Three times a week consisting of group aerobic exercise.	Walking groups	Personal trainers

Abbreviations: N = Number; % Female = Percentage of Female Participants; y = Years; SD = Standard Deviation; NR = Not Reported.

**Table 2 ijerph-21-00601-t002:** Risk of bias assessment—EPHPP scores of included studies.

Study Authors and Date	Selection Bias	Study Design	Confounders	Blinding	Data Collection Method	Withdrawals and Dropouts	Overall Quality Score
Kirwan et al., 2022 [14]	Moderate	Moderate	Moderate	Moderate	Strong	Weak	Moderate
Kirwan et al., 2021 [15]	Moderate	Moderate	Strong	Moderate	Strong	Weak	Moderate
Akinci et al., 2018 [22]	Moderate	Strong	Moderate	Strong	Strong	Moderate	Strong
Mendes et al., 2017 [23]	Moderate	Moderate	Strong	Moderate	Strong	Moderate	Strong
Higgs et al., 2016 [24]	Moderate	Moderate	Moderate	Moderate	Strong	Moderate	Strong
Mendes et al., 2016 [25]	Moderate	Moderate	strong	Moderate	Strong	moderate	Strong
Cugusi et al., 2015 [26]	Moderate	Moderate	Weak	Moderate	Strong	Strong	Moderate
Negri et al., 2010 [27]	Moderate	Moderate	Strong	Moderate	Strong	Strong	Strong

## Data Availability

Data can be made available upon request.

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
