# Peer review of "The Effectiveness of Clinician-Led Community-Based Group Exercise Interventions on Health Outcomes in Adults with Type 2 Diabetes Mellitus: A Systematic Review and Meta-Analysis"

_ijerph, 2024, doi:10.3390/ijerph21050601_

Round 1
Reviewer 1 Report
Comments and Suggestions for Authors
This systematic review and meta-analysis set out to evaluate the combination of clinician-led and community-based exercise interventions for adults with T2DM, while also investigating the effectiveness of community-based group exercise interventions on health outcomes. Although this review is intriguing and relevant, it falls short in providing sufficient evidence of rigor in the systematic search strategy and the results. This may have negative consequences for the overall results of your manuscript.
I have included some specific feedback for you to consider:
1. There is insufficient evidence of rigor and reproducibility in the systematic literature search. For example, there is inaccuracy in the search results. The PRISMA flow chart should show 9 final papers, and not 8. This may have consequences for the subsequent data analysis. There is insufficient evidence of a reproducible search string, which should include MeSH terms. For example, the manuscript should include copies of the database searches. Please address these issues. Also, consider updating PROSPERO with the database search results.
2. The PRISMA flow chart is not referenced in the citations.
3. It is unclear how the authors have applied the EPHPP framework to assess the quality of the included studies. In some instances, papers are assigned an overall quality score of strong despite not meeting the criterion of having at least four strong ratings out of the six components. Please provide additional information to justify your rating scores. Also, it is worth noting that the decision to use the EPHPP framework diverges from the initially planned use of the JBI tool, as stated in the PROSPERO registration. Why?
4. From the information detailed the final two paragraphs of the introduction and the specified systematic review, please provide a clear rationale for why conducting this systematic review and meta-analysis is justified.
5. In the introduction, please define what you mean when you say clinician-led (how are personal trainers’ clinicians?) and explain the difference between community-based exercise interventions and facility-based or individual approaches.
6. I note that you state this is the first systematic review targeting the efficacy of supervised group exercise interventions for adults with T2DM. To ensure accurate representations of your review, please detail how your systematic review provides insights into the effectiveness of clinician-led community-based interventions when compared to standard care without the specified clinician-led (as stated in PICO).
7. It would be helpful to include specific detail of the inclusion/exclusion criteria within your methods section 2.3. Please justify why studies were omitted is published prior to 2003 and define the types of studies included in the review. For example, randomised control trials etc. Also, consider including the data extraction spreadsheet detailed in section 2.4 as a supplementary file.
8. Table 1, within the participant characteristics, it would be helpful to the reader to include numbers of adults with T2DM for each study.
9. Within the results section, figures are no labelled and images of results are of a poor quality and not easy to read.
1 The abstract lacks crucial information from the review. For example, the methods section needs to clearly state the inclusion and exclusion criteria, the databases searched and the date the databases were last searched. Please refer to the PRISMA 2020 checklist for Abstracts.
1 In the introduction, cited references to support and strengthen your argument, particularly in relation to exercise and type 2 diabetes, may benefit from the inclusion of recent publications.
Comments on the Quality of English LanguageNot applicable.
Reviewer 2 Report
Comments and Suggestions for Authors
Title: The Effectiveness of Clinician-Led Community-Based Group Exercise Interventions on Health Outcomes in Adults with Type 2 Diabetes Mellitus: A Systematic Review and Meta Analysis
Reviewer Comments: The combined effects of community-based and clinician-led group exercise programs on a variety of health outcomes in individuals with Type 2 Diabetes Mellitus (T2DM) were assessed in this systematic review and meta-analysis. The study covered eight peer-reviewed trials with 938 participants and concentrated on how the therapies affected anthropometric and hematological measures, physical fitness, and glycemic management. Five studies were examined, and the results pertaining to physical fitness measured by the chair sit-and-reach test, the 30-second sit-to-stand test, and the six-minute walk test were all indicative of improvements. Seven studies' worth of anthropometric data showed improvements in diastolic blood pressure and waist circumference, but no discernible changes in body mass index, systolic blood pressure, weight, or resting heart rate. Reviewing four studies, hematological outcomes revealed significant improvements in total cholesterol, triglycerides, and fasting blood glucose. Glycemic management was demonstrated by a decrease in HbA1c levels. However, levels of LDL and HDL cholesterol did not change. Ten of the fifteen outcome measures that were evaluated shown a significant improvement, suggesting that the intervention techniques put in place may have a considerable positive impact on health when it comes to controlling important T2DM-related health parameters. These results, along with more research, may help to improve physical activity recommendations for people with type 2 diabetes, which would support community-based, supervised group exercise.
Strengths:
1. A large sample size of 938 participants improves the validity of the inferences made.
2. Integration of anthropometric, functional, and hematological variables provides a thorough understanding of the many effects on persons with type 2 diabetes.
Weaknesses:
1. There is no molecular data or experimentally derived data.
2. These types of studies are prone to bias, such as selection bias, attrition bias, and selective outcome reporting.
3. The figure on 187th page can be split. Figure is too big. Fonts are not visible.
4. It’s not clear whether these interventions can be applied to different geographical locations and different people.
5. What is the maximum age range at which participants in this study can be included?
6. To assess physical fitness and functional capacity, the following tests also would have mentioned. Ex: Exercise capacity testing, Incremental shuttle walk test, muscle strength testing.
7. IDL-C and Lp(a)-C levels also can be measured.
8. These PA interventions can be commonly applied to Males and Females?
9. Researchers combining various study types (apples and oranges) in one analysis is a common critique of meta-analyses. It is argued that the summary effect will overlook potentially significant variations amongst research.
Author Response
Thank you for taking the time to thoroughly review our paper. Please find our responses to your comments attached below. (for some reason the system has renamed the document).

Round 2
Reviewer 1 Report
Comments and Suggestions for Authors
Questions remain unanswered or addressed.
There remains insufficient evidence of rigor and reproducibility in the systematic literature search. The PRISMA flow chart remains incorrect. The updated PRISMA flow chart in version 2 of the manuscript has not been amended. The PRISMA flow chart shows 14 papers retained, less 5 is not 8! There remains insufficient evidence of a reproducible search string, which should include MeSH terms. The manuscript should include copies of the database searches. Please include supplementary evidence of a database search.
In the introduction the definition of a clinician has not been addressed. The original question remains unanswered. The authors have explained what facility-based interventions offer, but not how the interventions differ? To reduce confusion and support readers from other countries, please be specific and define the interventions, for example, what is a facility-based intervention discussed in the introduction. Also, within the introduction, explain your definition of a clinician.
Regarding my original question regarding table 1, I am unable to see this and therefore cannot confirm if the numbers of adults with T2DM for each study has been included. (Editors – please check).
The abstract still lacks crucial information from the review. In accordance with PRISMA guidance, the methods section needs to clearly state the inclusion and exclusion criteria, the databases searched and the date the databases were last searched.
Author Response
Thanks for your second review of our paper. Please find our responses to your comments below:
Reviewer comments |
Authors’ response |
There remains insufficient evidence of rigor and reproducibility in the systematic literature search. |
We hope the MeSH terms will assist with this. |
The PRISMA flow chart remains incorrect. The updated PRISMA flow chart in version 2 of the manuscript has not been amended. The PRISMA flow chart shows 14 papers retained, less 5 is not 8! |
Apologies for this, we neglected to upload the revised PRISMA- please now find included in the manuscript. |
There remains insufficient evidence of a reproducible search string, which should include MeSH terms. The manuscript should include copies of the database searches. Please include supplementary evidence of a database search. |
Thanks for this- we have attached a Supplementary file with all Mesh Terms and Search Strategy for each Database |
In the introduction the definition of a clinician has not been addressed. The original question remains unanswered. The authors have explained what facility-based interventions offer, but not how the interventions differ? To reduce confusion and support readers from other countries, please be specific and define the interventions, for example, what is a facility-based intervention discussed in the introduction. Also, within the introduction, explain your definition of a clinician. |
We have added a definition- Lines 88-90
We have added this detail: Lines 74-77 |
Regarding my original question regarding table 1, I am unable to see this and therefore cannot confirm if the numbers of adults with T2DM for each study has been included. |
Yes numbers have been included. |
The abstract still lacks crucial information from the review. In accordance with PRISMA guidance, the methods section needs to clearly state the inclusion and exclusion criteria, the databases searched and the date the databases were last searched. |
Thanks for your comments. Please see lines 15-19 in the abstract. |